# Stacking the odds for Golgi cisternal maturation

**Somya Mani, Mukund Thattai\***

Simons Centre for the Study of Living Machines, National Centre for Biological Sciences, Tata Institute of Fundamental Research, Bangalore, India

**Abstract** What is the minimal set of cell-biological ingredients needed to generate a Golgi apparatus? The compositions of eukaryotic organelles arise through a process of molecular exchange via vesicle traffic. Here we statistically sample tens of thousands of homeostatic vesicle traffic networks generated by realistic molecular rules governing vesicle budding and fusion. Remarkably, the plurality of these networks contain chains of compartments that undergo creation, compositional maturation, and dissipation, coupled by molecular recycling along retrograde vesicles. This motif precisely matches the cisternal maturation model of the Golgi, which was developed to explain many observed aspects of the eukaryotic secretory pathway. In our analysis cisternal maturation is a robust consequence of vesicle traffic homeostasis, independent of the underlying details of molecular interactions or spatial stacking. This architecture may have been exapted rather than selected for its role in the secretion of large cargo.

\*For correspondence: thattai@ncbs.res.in

**Competing interests:** The authors declare that no competing interests exist.

## Introduction

The cisternal maturation model is a hypothesis about how the Golgi apparatus works (*Emr et al., 2009*; *Luini, 2011*; *Glick and Luini, 2011*). It posits that secretory cargo travel in cisternal compartments that slowly mature from the *cis*-Golgi to the *trans*-Golgi composition. This is driven by three processes. (1) New *cis*-cisternae are created by the homotypic fusion of ER-derived COPII vesicles at the ER-Golgi intermediate compartment (ERGIC) (*Nakano and Luini, 2010*). (2) Old *trans*-cisternae dissipate by the vesiculation of the trans-Golgi network (TGN) into Golgi-to-surface carriers (*Bard and Malhotra, 2006*; *De Matteis and Luini, 2008*). (3) Golgi-resident proteins are recycled along retrograde vesicles, including COPI vesicles, moving from older to younger cisternae (*Papanikou et al., 2015*). Similar processes govern the maturation of endosomes in the endocytic pathway (*Rink et al., 2005*; *Poteryaev et al., 2010*; *Huotari and Helenius, 2011*).

Cisternal maturation allows cells to process and secrete large cargo (*Nakano and Luini, 2010*). However, large secretory cargo would be impossible without a fully developed cisternal maturation mechanism in the first place. Exaptation can resolve this evolutionary Catch-22 paradox: if a complex trait arose in stages through neutral or non-adaptive processes, it could later be co-opted for new cellular functions (*Lynch, 2007a*). Such a mechanism explains the origins of complex eukaryotic features such as splicing, transcriptional feedback loops, and multi-subunit protein complexes (*Lynch and Conery, 2003*; *Lynch, 2007b*; *Fernández and Lynch, 2011*). Here we extend this idea beyond genomes, transcriptomes and proteomes, to eukaryotic membrane organization: we demonstrate that cisternal maturation is a near-inevitable consequence of the basic processes of vesicle traffic and cellular homeostasis, prior to any selection for function. The architecture of the Golgi could therefore have arisen non-adaptively, enabling the subsequent development of large-cargo secretion. Our results refute the Catch-22 and provide a strong evolutionary foundation for the cisternal maturation model, reinforcing the wealth of cell-biological evidence in its favor across multiple cellular contexts (*Bonfanti et al., 1998*; *Losev et al., 2006*; *Matsuura-Tokita et al., 2006*).

## Results

### From molecular rules to vesicle traffic

We set out to understand how molecular specificity and molecular exchange combined to determine the structure of a vesicle traffic network. Endomembrane vesicles and compartments are dynamically generated and maintained through specific molecular interactions (*Munro, 2004*). Coats and adaptors drive vesicle cargo loading and budding (*Bonifacino and Lippincott-Schwartz, 2003*; *Robinson, 2004*; *Traub, 2009*). Tethers and SNAREs regulate vesicle transport and fusion (*Yu and Hughson, 2010*; *Jahn and Scheller, 2006*; *Wickner and Schekman, 2008*). GTPases of the Arf and Rab families coordinate these events via regulatory cascades (*D'Souza-Schorey and Chavrier, 2006*; *Stenmark, 2009*). Vesicle exchange necessarily causes changes to source and target compartments, whose compositions are determined as an outcome of molecular gain and loss. This feedback underpins the complexity of vesicle traffic networks.

Previous mathematical analyses of vesicle traffic have used stochastic frameworks or continuous differential equations, and included such details as compartment size, location and chemical composition (*Heinrich and Rapoport, 2005*; *Binder et al., 2009*; *Dmitrieff and Sens, 2011*; *Dmitrieff et al., 2013*; *Ramadas and Thattai, 2013*). These approaches require a large number of quantitative kinetic parameters to analyze even simple systems with a few molecules and compartments, and are not suited to explore cell-wide vesicle traffic networks. Here we are primarily interested in cell-wide homeostatic network topologies. We have previously shown (*Ramadas and Thattai, 2013*) that topological features of a vesicle traffic network – the number of compartments and their connectivity – are robustly determined by qualitative molecular specificity rather than quantitative kinetics. We therefore formalize the properties of vesicle traffic using a Boolean framework (*Mani and Thattai, 2016*): we approximate molecular specificities and compartment compositions by a series of 1 s and 0 s (Materials and methods: The Boolean vesicle traffic model). We assume the cytoplasm is well mixed, and do not consider spatial organization; we are agnostic to the relative amounts of molecules on each vesicle, the size of vesicles and compartments, and the quantitative kinetic flux of vesicles between compartments. Boolean models have been successfully applied to transcriptional and signaling networks (*Kauffman et al., 2003*; *Li et al., 2004*; *Chaves et al., 2005*); but have not previously been used to study vesicle traffic. Our Boolean approach allows us to efficiently sample a large space of cell-biological rules, and produces results that are qualitatively similar to more detailed microscopic vesicle traffic models (*Figure 1G,H*; *Box 1*).

We consider a cell to be a collection of compositionally distinct membrane-bound compartments exchanging vesicles. Suppose there are $N$ types of membrane-associated, transmembrane or lumenal molecules that determine the properties of compartments and vesicles. We refer to these as active molecular labels: they influence budding and fusion, and can be used to define the identity of compartments. The Boolean composition of a compartment or vesicle is given by a binary vector of length $N$: each element of the vector is 1 or 0 depending on whether that molecular label is present in high or low amounts. The state of a cell at any timepoint is specified by giving the list of compartments of various compositions it contains. A stream of vesicles bud out of source compartments and fuse into target compartments, as specified by budding and fusion matrices (*Figure 1A,D*). Over a timescale of minutes this flux depletes the source compartment and enriches the target compartment in the molecular labels carried by the vesicles. Compartment compositions are updated from one timepoint to the next by mass balance, applied to each molecular label (blue arrows, *Figure 1B, E*): if it's coming but not going, you eventually gain it; if it's going but not coming, you eventually lose it. Apart from homotypic vesicle fusion (brown arrows, *Figure 1B,E*) which we discuss below, this is the sum total of our model.

Though we refer to vesicle transport throughout the text, our Boolean framework can also accommodate non-vesicular pathways. Some types of molecular labels can exchange between compartments only via transport vesicles. These include most lipids, and transmembrane proteins such as receptors and SNAREs. Others can travel from one compartment to another via the cytoplasm, either directly or on non-vesicle carriers. These include some types of lipids via carrier proteins, as well as the Arf- and Rab-family GTPases which switch between soluble and membrane-associated forms (*Stenmark, 2009*). If we explicitly assign a label to membrane lipids, any pathway carrying a single non-membrane molecular label can optionally be interpreted as direct cytoplasmic transport.

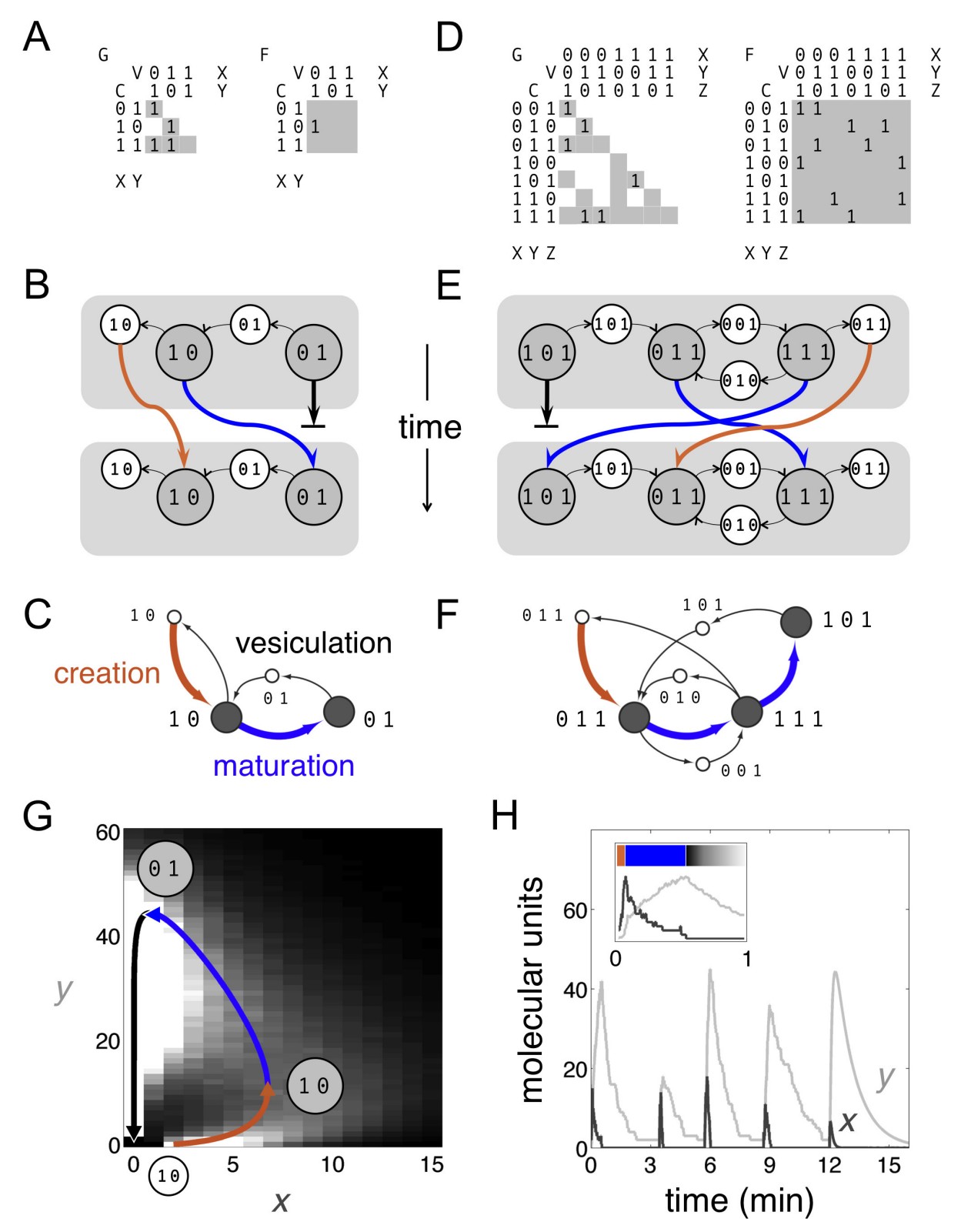

**Figure 1.** Boolean dynamics of compartments and vesicles: examples of maturation chains. (A,B,C) A system with $N = 2$ molecular labels $X$ and $Y$. (D, E,F) A system with $N = 3$ molecular labels $X, Y$ and $Z$. (A,D) Left: the $C \times V$ budding matrix $G$ : $G_{ij} = 1$ means compartment type $i$ (row) buds out vesicle type $j$ (column). Vesicle compositions are subsets of source compartment compositions, so only some entries of $G$ can be non-zero (gray shading). Right: the $C \times V$ fusion matrix $F$ : $F_{ij} = 1$ means vesicle type $j$ (column) can fuse with compartment type $i$ (row). Any entry of $F$ can be non-zero

*Figure 1 continued on next page*

*Figure 1 continued*

(gray shading). (B,E) At each timepoint compartments (gray circles) accept (through fusion) or give away (through budding) vesicles (white circles). Gray rectangles show the state of the cell at successive timepoints, moving from top to bottom. Compartments can undergo maturation (blue arrows) or vesiculate by giving up all their cargo (thick black arrows); orphan vesicles can undergo homotypic fusion to generate new compartments (brown arrows). From the first to the second timepoint, individual compartments change composition but the full set of compartments is constant: the system has reached homeostasis. (C,F) Another representation of the homeostatic networks from *Figure 1B,E*, mapping out a cell in compositional space. Circles represent compartments (gray) or vesicles (white); each distinct circle represents a distinct compositional type. Thin black arrows show vesicles moving between compartments at one timepoint. Brown arrows followed by blue arrows show the creation and maturation of compartments, flowing from one timepoint to the next. (G,H) Results of a stochastic simulation of vesicle traffic for a system with two molecular labels *X* and *Y* (Box, *Equation 1*). The cell contains many vesicles and compartments, and reaches an equilibrium in which the vesicle pools and the number of compartments of each compositional type are approximately constant. Parameters: $\{x_{tot}, y_{tot}\} = \{250 \text{ units}, 5000 \text{ units}\}$; $\{A, B, C, D\} = \{1000 \text{ units min}^{-1}, 10 \text{ min}^{-1}, 5 \text{ min}^{-1}, 1 \text{ min}^{-1}\}$; time-averaged vesicle pools $\bar{n}_X = 0.457$, $\bar{n}_Y = 10.6$. Timescales are chosen to qualitatively match real maturation dynamics (*Losev et al., 2006*). (G) The heatmap shows the equilibrium distribution of compartment compositions. Individual compartments change composition over time: the curve shows the deterministic limit-cycle solution to *Equation 2*, with different phases corresponding to creation (brown), maturation (blue), and vesiculation (black). Extrema of this curve correspond to the Boolean compartment compositions in *Figure 1C*. (H) Compositions of individual compartments over time for the full stochastic simulation (*X* in dark gray, *Y* in light gray). Compositions cycle periodically; each cycle is independent of the previous one, since new compartments are nucleated by fresh homotypic fusion events. The final cycle shows the deterministic limit-cycle solution to *Equation 2* for comparison. The inset expands the first minute, with *X* and *Y* levels scaled against their maximum values so creation, maturation and vesiculation can be clearly observed. *Figure 1—figure supplement 1* shows how non-vesicular transport can be included in this framework.

The following figure supplement is available for figure 1:

**Figure supplement 1.** Combining vesicular and non-vesicular transport.

Such an approach can be used, for example, to model Rab conversion during endosomal maturation (*Rink et al., 2005*) (*Figure 1—figure supplement 1*).

## Orphan vesicles and homotypic fusion

We assume a clear hierarchy of membrane-bound structures: compartments are large, transport vesicles are small. The behavior of vesicles and compartments depends on their compositions, but can also depend on their size, for example through the influence of curvature or membrane tension. Suppose the fusion matrix specifies that vesicles of composition *A* fuse to compartments of composition *B*. This does not automatically imply that two vesicles of compositions *A* and *B*, or two compartments of compositions *A* and *B*, must fuse to one another. Here we explicitly assume that compartments cannot fuse heterotypically to one another, and that transport vesicles cannot fuse heterotypically to one another. However, compartments (*Stenmark, 2009*) and transport vesicles (*Nakano and Luini, 2010*) are known to fuse homotypically in certain situations. Homotypic compartment fusion, if it does occur, has no influence on our Boolean dynamics; in contrast, homotypic vesicle fusion has a key role to play. The dynamics often generate orphan vesicles that find no target compartment. Left unchecked these would leak out of the system, preventing homeostasis. Here we focus on homeostatic states in which a small specific subset of orphan vesicles can undergo homotypic fusion and nucleate new compartments, as seen in real cells (*Nakano and Luini, 2010*). Such vesicles would have to fuse much more efficiently to one another than to compartments of the same composition, perhaps mediated by size-dependent effects (brown arrows, *Figure 1*; *Box 1*).

The following algorithm efficiently generates all homeostatic states with these properties (*Figure 1B,E*; Materials and methods: The Boolean vesicle traffic model). Starting from some initial condition, we run Boolean updates. If at any point we encounter a homeostatic state with no orphans, we stop. If we encounter a state with orphan vesicles, we check what happens if specifically those vesicles could homotypically fuse to nucleate compartments. If this produces a homeostatic state, we stop. If not, we use the resulting state as our initial condition. We nullify assumptions about homotypic vesicle fusion, and repeat the process. In practice we find that a small dose of homotypic fusion is often sufficient to achieve homeostasis: over 80% of our homeostatic networks require two or fewer types of homotypically fusing vesicles.

## Box 1. Stochastic, continuous, and Boolean models of cisternal maturation.

It is interesting to see how our Boolean model is related to more microscopic vesicle traffic models (*Figure 1*). Consider a system with two types of molecules *X* and *Y*, transported on distinct types of vesicles. We assume a well-mixed cytoplasm containing many compartments and vesicles. Each vesicle contains a fixed amount of each molecule, providing a measurement unit; anything larger than a vesicle is a compartment. We assume *X* vesicles can fuse to one another and, much less efficiently, to compartments; *Y* vesicles can fuse only to compartments; and compartments cannot fuse to one another. Let $n(x,y)$ be the number of compartments with $x$ units of *X* and $y$ units of *Y*. The number of free vesicles of each type is: $n_X \equiv n(1,0)$, $n_Y \equiv n(0,1)$. The total amount of each molecule remains constant, given by $x_{tot} = \sum_{x,y} n(x,y)x$, $y_{tot} = \sum_{x,y} n(x,y)y$. The allowed reactions are:

$$
\begin{aligned}
&\text{Two } X \text{ vesicles fuse homotypically to create a compartment:}\\
&n_X \downarrow_2, n(2,0) \uparrow^1 && \text{rate } An_X(n_X - 1)\\
&\text{A compartment buds an } X \text{ or } Y \text{ vesicle:}\\
&n_X \uparrow^1 \; n(x,y) \downarrow_1, n(x-1,y) \uparrow^1 && \text{rate } Bx\, n(x,y)\\
&n_Y \uparrow^1 \; n(x,y) \downarrow_1, n(x,y-1) \uparrow^1 && \text{rate } Dy\, n(x,y)\\
&\text{A compartment fuses to an } X \text{ or } Y \text{ vesicle:}\\
&n_X \downarrow_1 \; n(x,y) \downarrow_1, n(x+1,y) \uparrow^1 && \text{rate } A\frac{x^2}{(x+y)^2} n_X n(x,y)\\
&n_Y \downarrow_1 \; n(x,y) \downarrow_1, n(x,y+1) \uparrow^1 && \text{rate } Cx\, n_Y n(x,y)
\end{aligned}
\tag{1}
$$

Here, up/down arrows indicate increases/decreases in number, and rates correspond to reaction probabilities per unit time. Each term arises as follows: *X* or *Y* vesicles bud in proportion to the amount of *X* or *Y* on each compartment ($Bx$, $Dy$ terms). *X* promotes fusion of vesicles to compartments, cooperatively for *X* vesicles ($Ax^2$ term) and linearly for *Y* vesicles ($Cx$ term). The rate of fusion of *X* vesicles decreases sharply as the size of the compartment increases ($(x+y)^{-2}$ term); when a compartment is identical to an *X* vesicle this reduces to the homotypic fusion rate constant *A*. The parameters have the following dimensions: $n(x,y)$, $n_X$ and $n_Y$ are pure numbers of compartments and vesicles in the cell; *x* and *y* are measured in units corresponding to the amount of *X* or *Y* on a single transport vesicle; *A* has dimensions of molecular units per unit time; *B*, *C*, and *D* have dimensions of inverse time. The value $D^{-1}$ is the approximate time a compartment takes to fully vesiculate.

We simulate this stochastic system using Gillespie's algorithm (*Gillespie, 2007*). From arbitrary initial conditions the system reaches a homeostatic state in which the free vesicle pools and the number of compartments of any given composition are nearly constant (*Figure 1G*). However, individual compartment compositions vary over time. In the mean-field limit each compartment approximately obeys a continuous deterministic version of *Equation 1*, where $\bar{n}_X$ and $\bar{n}_Y$ are time-averaged values:

$$
\begin{aligned}
\frac{dx}{dt} &= A\bar{n}_X \frac{x^2}{(x+y)^2} - Bx\\
\frac{dy}{dt} &= C\bar{n}_Y x - Dy
\end{aligned}
\tag{2}
$$

By linear stability analysis, under the following parametric condition this system has no stable fixed point:

$$
\frac{B}{B+D} \frac{C\bar{n}_Y}{C\bar{n}_Y + D} > \frac{1}{2}
\tag{3}
$$

Roughly, *X* vesicles must bud rapidly and *Y* vesicles must fuse rapidly. Compartment compositions then approach a limit-cycle trajectory (*Figure 1G,H*): nucleated by homotypic fusion of *X* vesicles (brown); maturing by budding *X* vesicles and fusing with *Y* vesicles (blue); and finally vesiculating into *Y* vesicles (black). These dynamics are similar to live-cell observations of

cisternal maturation (*Losev et al., 2006*). *Figure 1A* shows the budding and fusion matrices corresponding to the Boolean version of *Equation 1*. The resulting Boolean homeostatic state (*Figure 1B,C*) displays the same qualitative features as the stochastic and continuous models (*Figure 1G,H*). These examples all describe the maturation chain, at different levels of abstraction.

## Statistical cell biology and non-adaptive evolution

In the most basic maturation chain, a compartment created by homotypic vesicle fusion matures into one or more successive compositional types, coupled with retrograde transport of cargo (brown creation arrow followed by blue maturation arrows, *Figure 1C,F*; *Box 1*). Multiple maturation chains could exist as an interlinked part of a larger vesicle traffic network (*Figure 2C*). If we precisely set the elements of the budding and fusion matrices by hand, we can ensure that vesicles of the right composition bud and fuse between the right compartments so as to generate maturation chains (as with the examples in *Figure 1*). But what if the elements of these matrices are set at random? What is the likelihood that complex structures such as maturation chains would then arise?

This question is relevant because existing genetic variation constrains the directions in which natural selection can operate. It is therefore important to identify structures that arise in vesicle traffic systems prior to any selection for function. Inspired by statistical physics, we do a form of statistical cell biology: we make as few assumptions as possible, and thus sample a diverse ensemble of vesicle traffic systems. Just as polling a representative subset of likely voters provides information on election outcomes, statistically sampling a representative subset of budding and fusion rules reveals properties intrinsic to vesicle traffic. This represents a neutral null hypothesis: if a structure of interest occurs frequently in this sample without any fine tuning, it is consistent with a non-adaptive evolutionary origin.

## Sampling homeostatic vesicle traffic networks

To systematically explore diverse behaviors, we sorted vesicle traffic systems according to the number of molecular labels, the number of adaptor/coatomer types, and how permissive vesicles were in selecting cargo and fusion targets (*Figure 2—figure supplement 1*; Materials and methods: Sampling homeostatic vesicle traffic networks). These parameters capture genetically-encoded properties such as the number, and the degree of interaction specificity, of cargo, adaptors, Arf and Rab GTPases, and SNAREs.

For each set of parameter values, we generated random instances of budding and fusion matrices (placed 1 s and 0 s at random positions). Given a budding and a fusion matrix, an arbitrary collection of compartments is almost certainly unstable: compartment compositions will update through molecular exchange, until finally the system settles into an unvarying homeostatic state (*Figure 1B, E*; *Figure 2A–C*). Different initial conditions lead to different homeostatic states. Each such state will typically contain only a small subset of all possible compartment compositions, and therefore depend only on a small number of elements of the full budding and fusion matrices. A homeostatic state represents a very special collection of compartments: either individual compartments in this set equalize gain and loss via vesicles – the balanced vesicular transport condition – or else different compartment types must interconvert between one another – the compartmental maturation condition.

Within a homeostatic state, compartments could segregate into subsets connected within themselves but disconnected from one another (*Figure 2—figure supplement 2A*). Most trivially, each compartment could be completely disconnected from all the others. In total we collected 63,897 homeostatic networks (35,838 non-trivial) for $N = 4, 5, 6, 7$ molecular label types. We found that network properties depended only weakly on $N$ (*Figure 2—figure supplement 1A,B*) so we focus the rest of the discussion on the 14,809 homeostatic networks (8614 non-trivial) obtained for $N = 7$ label types. For this number of molecular labels, there are $2^7 - 1 = 127$ possible compartment or vesicle

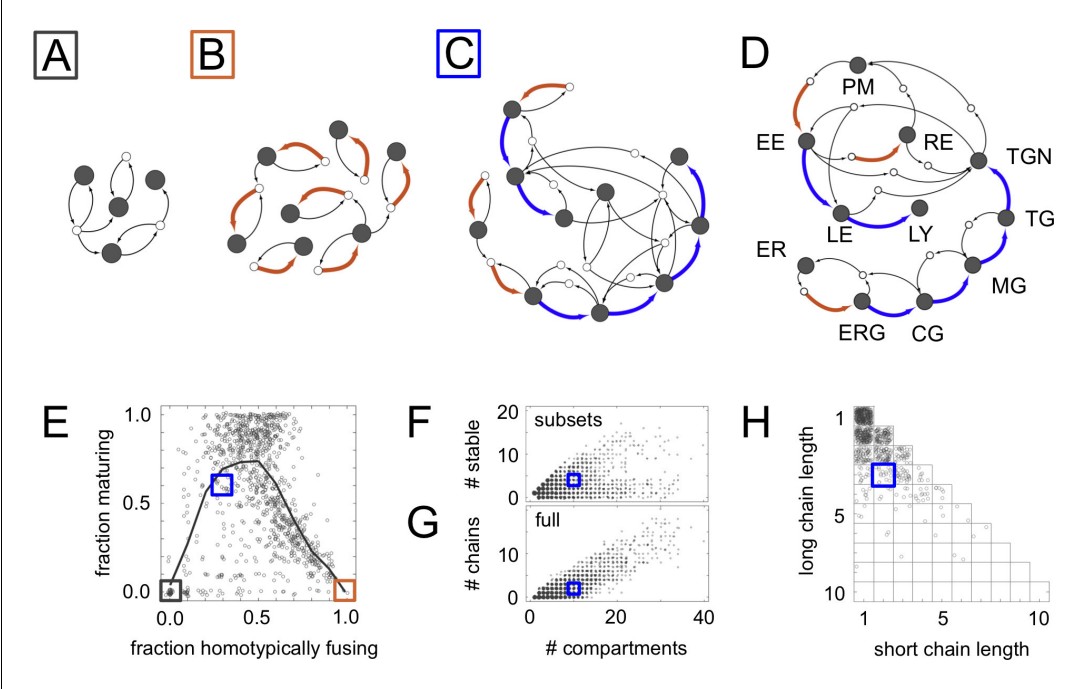

**Figure 2.** Homeostatic vesicle traffic networks. These data relate to the 14,809 homeostatic networks for $N = 7$ molecular label types. (**A–C**) Examples of homeostatic networks (compartment and vesicle compositions are omitted for clarity) for budding and fusion rules generated using different parameter values $\{N, A, f, g\}$ (Materials and methods: Sampling homeostatic vesicle traffic networks). (**A**) {7, 1, 0.35, 0.35}, gray square icon. This is a transport-balanced network with no homotypic fusion or maturation. (**B**) {7, 4, 0.025, 0.75}, brown square icon. This is a network with high homotypic fusion but no maturation, and is broken into many disconnected subsets. (**C**) {7, 2, 0.1, 0.7}, blue square icon. This network has low homotypic fusion and high maturation, similar to a real eukaryotic cell. (**D**) Schematic vesicle traffic system of a eukaryotic cell. ER: endoplasmic reticulum. ERG: ER-Golgi intermediate compartment. CG: *cis*-Golgi. MG: medial-Golgi. TG: *trans*-Golgi. TGN: *trans*-Golgi network. PM: plasma membrane. EE: early endosome. LE: late endosome. LY: lysosome. RE: recycling endosome. (**E**) For each network in our dataset with ten or more compartments, we show the fraction of compartments undergoing maturation (outgoing blue arrow) vs. the fraction of compartments created by homotypic vesicle fusion (incoming brown arrow). The gray curve shows a moving average of the data. The colored squares show the position of the three networks in *Figure 2A–C*. Maturation is most likely at low-to-moderate rates of homotypic fusion. (**F**) The number of stable compartments vs. the total number of compartments for all the connected subsets in our dataset. The ER and plasma membrane are examples of stable compartments in real cells. (**G**) The number of maturation chains vs. the total number of compartments for all the networks in our dataset. (**H**) Chain lengths for all 841 networks in our dataset with precisely two maturation chains. In (**E–H**) we plot values with a random additive noise so the density of points can be observed. The blue square icon shows the properties of the cell-like network from *Figure 2C*. *Figure 2—figure supplements 1,2* explore further properties of these homeostatic networks.

The following figure supplements are available for figure 2:

**Figure supplement 1.** Statistical sampling of networks.

**Figure supplement 2.** Properties of homeostatic networks.

compositions, so the budding and fusion matrices contain thousands of elements (*Figure 2—figure supplement 2F,G*).

Among the networks generated by this procedure, 90% had 8 or fewer compartments and 11 or fewer vesicle types (*Figure 2—figure supplement 2B,C*). The compartments showed a bimodal distribution of compositional complexity, with an excess of compartments having very many or very few molecular labels (*Figure 2—figure supplement 2D*, top). In contrast, vesicles tended to be compositionally simple (*Figure 2—figure supplement 2D*, bottom). Molecular loss was offset by molecular gain at each stable compartment on fast timescales. However, compartment groups sometimes collectively received molecules that remained trapped within the group. Such molecular sinks create a need for compensatory synthesis elsewhere on slower timescales. We found that larger networks required synthesis of more types of molecular components (*Figure 2—figure supplement 2E,H*).

## Cell-like networks and vesicle traffic motifs

A significant fraction of our homeostatic networks bore a striking resemblance to real eukaryotic vesicle traffic networks (*Figure 2*). In particular, they contained maturation chains. In terms of the number of compartments, number of chains, and chain lengths, the real eukaryotic traffic network (*Figure 2D*) appeared to be a typical example of a homeostatic network generated by our procedure (*Figure 2C,E–H*). Cell-like networks containing maturation chains were most likely to occur at low-to-moderate rates of homotypic vesicle fusion (*Figure 2E*): at very low rates all networks were of the transport balance type (*Figure 2A*); at very high rates they broke into many disconnected vesiculating subsets (*Figure 2B*). Nearly half the non-trivial homeostatic networks (4024/8614) contained a maturation chain of length one or more.

Unexpectedly, we found that the lengths of maturation chains were geometrically distributed (*Figure 3C*). Such distributions typically arise for processes in which each step is random and independent, for example counting the number of heads before hitting the first tail in a series of coin flips. This suggests that vesicle traffic networks might usefully be described by the random graph generation approaches that have provided insights into the structure of metabolic, neural, and ecological networks (*Albert and Barabási, 2002*) (Materials and methods: Randomly shuffled vesicle traffic networks).

We wondered whether the networks lacking maturation chains were characterized by other distinct features. To quantify this, we did an unbiased search for highly represented network motifs

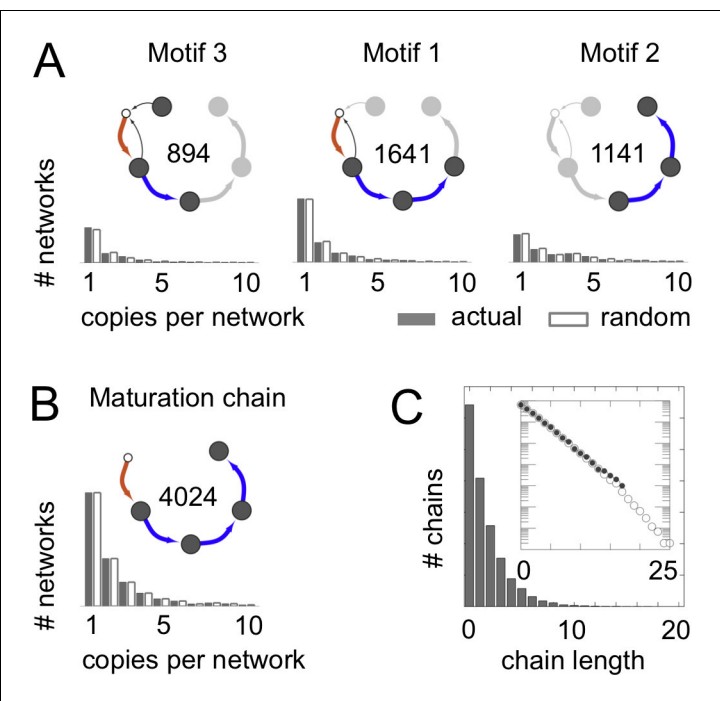

**Figure 3.** Frequent vesicle traffic motifs. These data relate to the 14,809 homeostatic networks for $N = 7$ molecular label types. (**A,B**) Insets indicate the number of homeostatic networks that contain at least one copy of the motif. Histograms show the number of repeated copies of each motif per network, in actual (gray bars) or randomized (white bars) networks (Materials and methods: Vesicle traffic motifs; Materials and methods: Randomly shuffled vesicle traffic networks). (**A**) The three most frequent connected three-compartment motifs. (**B**) The maturation chain. (**C**) Histogram of maturation chain lengths. We consider any compartment with an incoming brown arrow as the start of a chain, and define length as the number of subsequent blue arrows. Inset: the same data in a log-linear plot. Both the real (grey dots) and randomized (white dots) networks show a geometric distribution of chain lengths. *Figure 3—figure supplement 1* shows the frequency of the top 100 most common motifs.

The following figure supplement is available for figure 3:

**Figure supplement 1.** Vesicle traffic motif statistics.

(*Milo et al., 2002*) (*Figure 3*; *Figure 3—figure supplement 1*; Materials and methods: Vesicle traffic motifs). There are 43,700 ways in which three compartments can be connected to one another to form a motif. Strikingly, the three most frequent three-compartment motifs (*Figure 3A*) were all subgraphs of the maturation chain, and eight of the ten most frequent motifs themselves contained maturation chains (*Figure 3—figure supplement 1*). In contrast, the 4590 non-trivial networks with no maturation chains showed no clear pattern: they were characterized by about 50 different motifs, no single one of which was found in more than 500 networks (*Figure 3—figure supplement 1*). The 15th most common three-compartment motif, occurring in 344 networks, is the maturation cycle (*Figure 3—figure supplement 1*). This corresponds to a compartment whose composition oscillates periodically. Cycles of various periods occurred in about a quarter (2407/8614) of all homeostatic networks. Out of 3173 non-trivial networks with no homotypic fusion, over a third contained cycles of maturing compartments (1053/3173).

## Retrograde vesicles emerge spontaneously, driving cisternal maturation

What is the fate of a compartment that is created by homotypic vesicle fusion (*Figure 4A*)? It can proceed through steps of maturation, and finally end up as either a compartment of fixed composition, or one whose composition oscillates over time. Most maturation chains (72%) terminated at fixed compartments; terminal oscillating compartments were uncommon (*Figure 4B*). The majority (61%) of such fixed terminal compartments gave up all their cargo as vesicles at the next timepoint, and thus dissipated. When we examined the flow of molecules and vesicles within maturation chains (*Figure 4C*), we found a striking and unexpected feature: younger compartments were disproportionately likely to receive retrograde vesicles directly from their older successors, over eight times more often than expected by chance (*Figure 4C*, right panel). Individual molecules treadmilled in place within a chain, hopping to younger compartments via retrograde vesicles, driving their conversion to the older composition. The first compartment of a maturation chain tended to be compositionally simpler than its successors (at least in terms of the active labels considered here) due to the treadmilling molecules present on the latter.

Distilling these observations, here is our central result: across 8614 non-trivial homeostatic networks, we find 3111 maturation chains which precisely match all three diagnostic features of the cisternal maturation model (*Figures 1,4*). (1) The first compartment is created by homotypic vesicle fusion and then matures via one or more steps. (2) The terminal compartment dissipates by vesiculation. (3) Younger compartments receive retrograde vesicles directly from their older successors.

## Discussion

We have answered the question posed at the outset: even when elements of the budding and fusion matrices are set at random, the stringent requirements of homeostasis drive cells toward highly structured compartment compositions, among which cisternal maturation chains frequently occur. The basic, well-established dynamics of vesicle traffic – in which the specificity of vesicle formation and fusion is locally regulated through molecular interactions – are sufficient to generate a Golgi-like structure without fine tuning. We emphasize that our results pertain to cell-wide vesicle traffic rather than narrowly to the Golgi apparatus, and have implications for maturation dynamics observed in other secretory and endocytic systems (*Mani and Thattai, 2016*). Almost all networks with maturation chains also contained stable compartments with balanced vesicular transport: these structures co-exist, they not mutually exclusive. Classic models of membrane traffic focus on the generation of stable compartments out of the dynamic process of vesicle exchange (*Heinrich and Rapoport, 2005*). The few that consider maturation introduce it by hand, and study subsequent features of protein transport (*Glick et al., 1997*; *Weiss and Nilsson, 2000*; *Dmitrieff et al., 2013*; *Ispolatov and Müsch, 2013*). In contrast, maturation is not built into our framework, it emerges spontaneously. When a basic model is sufficient to generate cisternal maturation, the burden of proof is on complex models to justify additional ingredients.

Our strongest result is the emergence of retrograde vesicles between maturing cisternae in the absence of a spatial coordinate. All earlier mathematical analyses explicitly or implicitly assumed that maturation was driven by retrograde vesicles moving between spatially neighboring compartments (*Glick et al., 1997*; *Weiss and Nilsson, 2000*; *Dmitrieff et al., 2013*; *Ispolatov and Müsch, 2013*). However, un-stacked Golgi are widespread (*Mowbrey and Dacks, 2009*); and do have retrograde

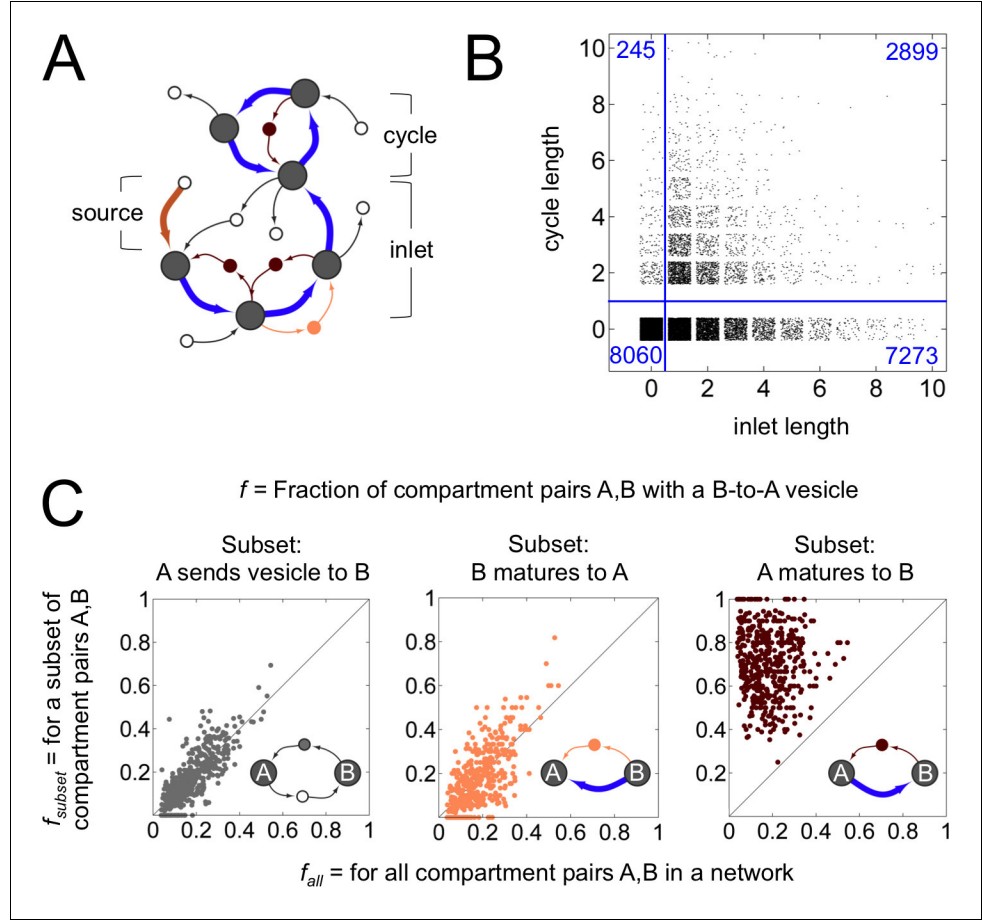

**Figure 4.** Anatomy of a cisternal maturation chain. (**A**) A compartment created at the first timepoint (source, thick brown arrow) matures at each successive timepoint (inlet, thick blue arrows). This process can terminate at a compartment with fixed composition, or one with an oscillating composition (cycle of thick blue arrows). Vesicles move molecules between compartments. We classify vesicles as anterograde (thin light brown arrows), retrograde (thin dark brown arrows) and other (thin black arrows). Some vesicles are inbound or outbound from other parts of the network. (**B**) Properties of the terminal compartment of a maturation chain. There are 10,172 homotypically fusing sources in our dataset that have at least one maturation step, of which just 2899 terminate in a cycle. The remaining 7273 sources terminate at 4588 fixed compartments (the reduction in number is because maturation chains can converge). Finally, 2820 of these 4588 compartments dissipate at the next timestep, giving up all their cargo as vesicles. (**C**) Fraction of compartment pairs A,B with a B-to-A vesicle. Each dot in each panel represents values for a single network; we show data only for networks that have ten or more maturation edges. The x-axis is the vesicle fraction $f_{all}$ for all compartment pairs A,B in the network. The y-axis is the vesicle fraction $f_{subset}$ for the subset of compartment pairs A,B in the network where A sends a vesicle to B (left), B matures to A (middle), or A matures to B (right). For the left and middle panels, the chance of finding a B to A vesicle for the corresponding subset of pairs is the same as the chance between all pairs in a network. For the right panel, the chance of finding a B to A vesicle when A matures to B (median 0.75) is eight fold more than the chance across all pairs (median 0.09). For example, suppose a network with 10 compartments has $10 \times 9 = 90$ compartment pairs A,B. Suppose for 8 of these pairs we know that A matures to B. Now we count vesicles. We find that 8/90 = 0.09 of all pairs and 6/8 = 0.75 of the maturing pairs have a vesicle going from B to A. This corresponds to a network with an x-value of 0.09 and a y-value of 0.75: there are many more retrograde vesicles between maturing compartments than expected by chance. That is, a younger compartment in a maturation chain almost certainly receives a retrograde vesicle from its immediate older successor. *Figure 4—figure supplement 1* explores the optimal spatial distribution of compartments in a maturation chain.

The following figure supplement is available for figure 4:

**Figure supplement 1.** Spatial optimization of networks.

vesicles (*Papanikou et al., 2015*). This is consistent with our results: it shows that spatial organization is not a pre-requisite for maturation, and supports the idea of a well-mixed cytoplasm.

A eukaryotic cell is an object distributed in both chemical and physical space (*Misteli, 2001*). The connectivity of a vesicle traffic network is primarily determined by molecular chemical specificity, while the locations of compartments are determined by complex and largely unknown biophysical mechanisms (*Yadav and Linstedt, 2011*). The spatial organization of a cell clearly impacts all aspects of its activity, including vesicle traffic. To add a physical dimension to our analysis, we explored the hypothesis that a cell is like an efficient city-wide logistics system: one in which no part of a city is too far from a warehouse, and warehouses with more traffic between them are placed closer to one another (*Taniguchi et al., 1999*). For each network in our dataset, we ran an optimization algorithm to distribute compartments uniformly across a cell while minimizing the distance travelled by vesicles between compartments (Materials and methods: Spatial optimization). Remarkably, maturation chains in such optimized networks immediately organized into spatially contiguous stacks: each compartment was adjacent in space to the type of compartment it matures to in time (*Figure 4—figure supplement 1*). This was mainly due to the excess of retrograde vesicles within maturation chains, though most vesicles still traveled long distances. The stacked morphology of the Golgi apparatus in many eukaryotes is thus consistent with the expectations of efficient traffic. Experiments with repositioned organelles might be used to test these ideas further (*van Bergeijk et al., 2015*). However, hypotheses about function and efficiency are much more difficult to prove than those about molecular mechanism. Our main result is based only on the latter: as vesicle traffic networks evolve, cisternal maturation is easily discoverable by mutation, prior to any selection. For the Golgi apparatus, form might have preceded function.

## Materials and methods

### The Boolean vesicle traffic model

We assume $N$ types of active molecular labels, and $A$ types of cytoplasmic adaptor/coatomer complexes. Compartments ($C$) and vesicles ($V$) are both collections of molecular labels, represented by binary row vectors of length $N$. Each element of such a vector is 1 or 0, indicating whether a certain label type is present in high or low amounts, loosely referred to as presence or absence. For example, for $N = 3$ the row vector [101] indicates that only the first and third label types are present on the corresponding compartment or vesicle. There are $C = 2^N - 1$ and $V = 2^N - 1$ possible non-zero types of compartments and vesicles, respectively. Vesicles with a single non-zero label may optionally be interpreted as membrane-associated molecules capable of cytoplasmic non-vesicular transport (*Figure 1—figure supplement 1*). Since our model is deterministic, a set of compartments of the same composition present at the same time will all have the same dynamics. It is therefore sufficient to keep track of whether some compartment of a given composition is present or absent. Each of the $\sim 2^N$ possible compartments could be present or absent at any timepoint, so the cell itself could be in one of $\sim 2^{2^N}$ possible states.

Let $i = 1, \ldots, C$ index the possible compartment composition vectors, and $j = 1, \ldots, V$ index the possible vesicle composition vectors. We use a convention in which vectors are sorted according to the standard binary ordering. Thus the row vector $c(i)$ or $v(j)$ has entries corresponding to the digits of the binary number $i$ or $j$. For example, for $N = 3$ the compartment composition corresponding to the index $i = 5$ is the row vector $c(5) = [101]$. If $n = 1, \ldots, N$ runs over the label types, $c_n(i)$ or $v_n(j)$ represents the $n$th component of the compartment or vesicle row vector.

The dynamics are entirely specified by two matrices (*Figure 1A,D*): the $C \times V$ budding matrix $G$, and the $C \times V$ fusion matrix $F$. If a budding element $G_{ij} = 1$, compartment type $i$ generates vesicle type $j$. Since any vesicle vector must be a subset of the source compartment vector, the set of allowed non-zero elements of $G$ has the form of a Sierpinski triangle. Each row of $G$ can have at most $A$ non-zero elements, since this is the number of distinct adaptor/coatomer complexes. If a fusion element $F_{ij} = 1$, vesicle type $j$ fuses into compartment type $i$, as long as the latter is present at that timepoint. Any element of $F$ can be non-zero. Under these row-column conventions the product $(GF')_{i_1 i_2}$ (mnemonic buddinG x Fusion') represents the compartment-to-compartment matrix whose entries give the number of distinct vesicle types coupling any source compartment ($i_1$) with any target compartment ($i_2$).

At any timepoint $t$, let the sorted set of distinct compartments be $I^t \equiv \{i_1^t, i_2^t, \ldots\}$, where $i_1^t < i_2^t < \ldots \leq C$. The budding matrix $G$ then specifies the sorted set of distinct vesicles being generated at that timepoint $J^t \equiv \{j_1^t, j_2^t, \ldots\}$, where $j_1^t < j_2^t < \ldots \leq V$. That is, $J^t = \{j \,|\, \exists i \in I^t \text{ s.t. } G_{ij} = 1\}$. For any compartment of interest $i \in I^t$ we can use $G$ to find the vesicles $J(i \rightarrow) \subset J^t$ budding out of it, and use $F$ to find the vesicles $J(i \leftarrow) \subset J^t$ fusing into it: $J(i \rightarrow) = \{j \in J^t \,|\, G_{ij} = 1\}$ and $J(i \leftarrow) = \{j \in J^t \,|\, F_{ij} = 1\}$. Note that $J^t = \cup_i J(i \rightarrow)$ but $J^t \supset \cup_i J(i \leftarrow)$ since some orphan vesicles might not have targets at that timepoint.

Compartments change composition discretely and synchronously, using a binary version of mass balance (**Figure 1B,E**). Consider the $k$th entry in the sorted list of compartments present at timepoint $t$: it has index $i_k^t \in I^t$. The shorthands

$$
\begin{aligned}
v_n(i_k^t \rightarrow) &= \ OR\{v_n(j) \,|\, j \in J(i_k^t \rightarrow)\} \\
v_n(i_k^t \leftarrow) &= \ OR\{v_n(j) \,|\, j \in J(i_k^t \leftarrow)\}
\end{aligned}
\tag{4}
$$

represent the bitwise OR of the vesicle row vectors in these sets, effectively collapsing them into a single outgoing and a single incoming vesicle row vector for the $k$th compartment. The composition of this compartment updates according to the following Boolean expression:

$$
c_n(i_k^{t+1}) \equiv c_n^{t+1} = v_n(i_k^t \leftarrow) \vee \left[\neg v_n(i_k^t \rightarrow) \wedge c_n(i_k^t)\right]
\tag{5}
$$

where the leftmost relationship defines the index $i_k^{t+1}$ as the binary number whose digits are given by the row vector $c_n^{t+1}$. Put simply, a compartment will gain a molecular label type if it is fusing in and not budding out, and will lose a molecular label type if it is budding out and not fusing in. For all other cases compartment composition does not change.

If at any timepoint some orphan vesicle type does not fuse to any of the available compartments, this cannot represent a homeostatic state. We then check what happens if each orphan type is separately allowed to undergo homotypic fusion to create a compartment (brown arrows, **Figure 1B,E**). In this way we generate a new set of compartments

$$
I^{t+1} = \underbrace{\{j \in J^t \,|\, j \notin \cup_k J(i_k^t \leftarrow)\}}_{\text{created compartments}} \cup \underbrace{\{i_1^{t+1}, i_2^{t+1}, \ldots\}}_{\text{maturing compartments}}
\tag{6}
$$

where any duplicates are removed and the indices are sorted. Any compartment which vesiculates by losing all its molecules updates to the zero compartment $c(0) = [000\ldots]$ and is not included in this set. The number of distinct compartments could increase through creation. It could decrease if any compartment loses all its molecules, or if two compartments at timepoint $t$ update to the same composition at timepoint $t+1$. If the new updated state is identical to the previous one, this indicates we have found a homeostatic state, and we stop the updates. Otherwise, we nullify assumptions about homotypic vesicle fusion, and repeat the updates. Homeostatic states generated in this way contain compartments identical in composition to each orphan vesicle. Since orphans do not fuse to any existing compartment according to the fusion matrix, they necessarily cannot fuse heterotypically to one another.

## Sampling homeostatic vesicle traffic networks

We sample a large number of budding and fusion matrices and examine the traffic networks that result. We generate these matrices using four parameters (**Figure 2—figure supplement 1**): $N$ is the number of distinct molecular label types; $A$ is the number of adaptor/coatomer complexes. The continuous parameter $g \in [0, 1]$ sets the propensity of molecular cargo loading (higher values of $g$ mean vesicles have more 1 s). The continuous parameter $f \in [0, 1]$ sets the propensity of fusion (higher values of $f$ mean the fusion matrix has more 1 s). All these parameters are ultimately genetically determined: $N$ and $A$ represent the number of protein types involved, while $g$ and $f$ summarize their interaction properties. For example, $g$ depends on cargo-adaptor specificity, and $f$ depends on SNARE-SNARE specificity. Our search is unbiased in the sense that we sample uniformly over these parameters.

We generate $G$ and $F$ as follows. Let $X \sim Ber(p)$ be a Bernoulli random variable, so $\wp(X = 1) = p$, $\wp(X = 0) = 1 - p$. For every possible compartment $i$ we define $A$ vesicle row vectors of length $N$, one

for each possible adaptor/coatomer: $v_n^a(i \rightarrow) = c_n(i) \cdot X$, where $X \sim Ber(g)$ and $a = 1, \ldots, A$. That is, each vesicle typically loads a fraction $g$ of the cargo types on source compartments. The budding matrix is then given by $G_{ij} = 1 \Leftrightarrow \exists a \, s.t. \, v(j) = v^a(i \rightarrow)$. Each row of $G$ can have anywhere from 0 to $A$ non-zero entries, since not all vesicles budding from each compartment are distinct, and the zero vesicle $v(0) = [000\ldots]$ is ignored. The fusion matrix is given by $F_{ij} = X$, $X \sim Ber(f)$. That is, each vesicle typically fuses to a fraction $f$ of all compartment types. Once $G$ and $F$ are constructed, they remain constant throughout the dynamics.

We scan parameter values $N \in \{4, 5, 6, 7\}$, $A \in \{1, 2, 3, 4, 5\}$, $g, f \in \{0.05, 0.10, \ldots, 0.90, 0.95\}$, and additionally $f \in \{0.025, 0.075, 1\}$, giving 8360 distinct combinations (*Figure 2—figure supplement 1*). For each parameter combination we sample ten $F$, $G$ rules as described above. For each rule we start with a random initial condition $I^{t=0} = \{i_1^0, i_2^0, \ldots\}$ generated by uniformly sampling between 1 and $2^N - 1$ initial compartments, each of which can uniformly take any of the $2^N - 1$ possible Boolean indices. We use the prescription described in *Equation 6* to update the system in discrete timesteps starting from the initial condition. Since this is a deterministic dynamical system over a discrete state space, once sufficient time has elapsed all trajectories will converge onto one or more periodic orbits: a set of states which form a cycle under updates. Each rule will in general have many periodic orbits as attractors, and different initial conditions can converge to different orbits. For each initial condition, we run the system for 1024 timesteps or until it reaches a periodic orbit. A period-1 orbit, which updates to itself, is particularly relevant: it is a homeostatic state (*Figure 1B,E*). We store all such homeostatic vesicle traffic networks for further analysis. A trivial state is one in which no compartment connects to any other. In our simulations for $N = 7$, 98% of initial conditions reached an orbit within 1024 steps, of which 72% were homeostatic states. In total we found 17,458, 16,248, 15,382, and 14,809 homeostatic states (of which 9456, 8990, 8778, and 8614 were non-trivial) for $N = 4, 5, 6, 7$, respectively.

## Vesicle traffic motifs

In a homeostatic steady state, the full set of compartments remains constant, though individual compartments can interconvert between one another. We can represent each steady state as a graph (*Figure 1C,F*), where nodes $\{i_1, i_2, \ldots\}$ are compartments, and there are three types of edges: vesicle edges, creation edges, and maturation edges. Vesicle edges (black) go from source to target compartments: $i_1 \rightarrow i_2$ wherever $(GF')_{i_1 i_2} = 1$. Creation edges (brown) connect source compartments to newly created compartments generated by homotypic fusion of orphan vesicles (*Equation 6*): $i \rightarrow j$ for each vesicle such that $j \in J(i \rightarrow)$, $j \notin \cup_k J(i_k^t \leftarrow)$. Maturation edges (blue) connect compartment compositions at two successive timepoints (*Equation 5*): $i_k^t \rightarrow i_k^{t+1}$, unless the compartment is in transport balance such that $i_k^t = i_k^{t+1}$. A compartment is considered to be involved in a maturation step if it has an outgoing maturation edge. Considering only creation and maturation edges, there are 43,700 distinct connected three-compartment motifs. A maturation chain starts at a compartment with an incoming creation edge, and proceeds via maturation edges, stopping at a compartment with no outgoing maturation edge. If a compartment encountered earlier in the chain is repeated, this indicates a cycle so the chain is terminated before the repeat, and the cycle is stored for further analysis (*Figure 4A,B*).

## Randomly shuffled vesicle traffic networks

One way to generate a random graph is to preserve the number and properties of individual nodes, but otherwise connect them at random (*Albert and Barabási, 2002*). For each homeostatic network in our dataset, we break the network into individual compartments, then reconnect them as follows. We remove all vesicle edges. We then randomly swap remaining edge targets, allowing only swaps between maturation edges or between creation edges. We reject the swap if: a self edge is lost; a self edge is created; or multiple edges of the same type arise between the same source and target nodes. This procedure preserves all individual node properties: self edges, and the separate indegrees and outdegrees of creation and maturation edges. We generate 1000 randomly shuffled networks for each original network. We find that these shuffled networks have precisely the same distribution of motif frequencies and maturation chain lengths as the original networks (*Figure 3A, BC*). This supports the idea that a random graph generation approach might capture relevant features of vesicle traffic networks.

## Spatial optimization

We want to distribute the compartments of a vesicle traffic network in such a way that all parts of a cell are close to at least one compartment, while the distance travelled by vesicles between compartments is minimized (*Figure 4—figure supplement 1*). Let $r_i$ give the vector position of each compartment $i$, so pairwise distances between compartments are $s_{ij} = |r_i - r_j|$. Define an energy function:

$$E = \sum_{i,j} \frac{1}{s_{ij}} + (a_0 + a_{ij})s_{ij}^2 \tag{7}$$

where $a_{ij} = 1$ if a vesicle goes from compartment $i$ to compartment $j$, and $a_0 = 0.1$ sets a baseline value. By varying compartment positions to minimize this function, we achieve the desired optimal configuration. Intuitively: compartments act as repelling charges and spread out as uniformly as possible, while vesicle fluxes act as attractive springs and try to be as short as possible. Starting from random initial conditions, we numerically determined the optimal minimum energy compartment positions in two dimensions.

## Acknowledgements

MT was supported in part by a Wellcome Trust-DBT India Alliance Intermediate Fellowship (500103/Z/09/Z). We thank Joel Dacks, Anjali Jaiman, Ramya Purkanti, and Madan Rao for useful discussions.

## Additional information

### Funding

| Funder | Grant reference number | Author |
|---|---|---|
| Wellcome Trust-DBT India Alliance | Intermediate Fellowship 500103/Z/09/Z | Mukund Thattai |

The funders had no role in study design, data collection and interpretation, or the decision to submit the work for publication.

### Author contributions

SM, Acquisition of data, Analysis and interpretation of data; MT, Conception and design, Analysis and interpretation of data, Drafting or revising the article

### Author ORCIDs

Mukund Thattai, http://orcid.org/0000-0002-2558-6517

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
