## [Decision Letter]

Thank you for submitting your article "Stacking the odds for Golgi cisternal maturation" for consideration by *eLife*. Your article has been reviewed by three peer reviewers, one of whom, Benjamin Glick, also guest edited the paper, and the evaluation has been overseen by Randy Schekman as the Senior Editor. The following individuals involved in review of your submission have agreed to reveal their identity: Alberto Luini (Reviewer #2).

The reviewers have discussed the reviews with one another and the Reviewing Editor has drafted this decision to help you prepare a revised submission.

Summary:

This paper explores the dynamic organization of the Golgi by a theoretical modeling of Boolean networks. The simplicity of the approach enables the authors to perform an unbiased survey of a large number of possible vesicle trafficking scenarios. A surprising and striking result is that many of the networks display the properties of compartmental maturation. Thus, the evolution of Golgi maturation can be viewed as a natural outcome of the intrinsic properties of vesicle trafficking systems. According to this view, the transport of large secretory cargo molecules was probably an adaptation that took advantage of an existing process.

Essential revisions:

The reviewers and reviewing editor had an interesting discussion about your paper. Multiple concerns were raised, and initially it seemed likely that the decision would be to decline the submission. In the end, we agreed that the story is interesting enough to merit a request for resubmission after major revisions. There is no guarantee that the revised version will pass muster, so you will need to decide whether the effort is worthwhile.

There is general agreement that the idea of approaching this problem with Boolean networks is creative and novel, and that the results are an important step forward if the approach is sound. But the following issues need to be addressed.

1) Your model assumes that compartment identity is determined by transmembrane and luminal components, which are necessarily exchanged by means of transport vesicles. But in reality, components recruited from the cytosol, such as GTPases and adaptors/coats and phosphoinositide modifying enzymes, play an important and perhaps central role in defining compartment identity. In some cases, Rab cascades are thought to regulate compartmental maturation. While your model necessarily is necessarily a simplified representation of reality, overlooking this fundamental property of the endomembrane system seems hard to justify.

2) The focus of your analysis is on the subset of networks that include maturation chains. But the majority of the networks do not show maturation. The neglect of those other outcomes is surprising. For example, how often is the traditional vesicle shuttle model replicated? What can be learned from analyzing the full set of networks?

3) The assumption that vesicles unable to find a target undergo homotypic fusion is troubling. This process may reflect the actual behavior of some vesicle types in cells, but why is homotypic fusion the default assumption? Does this feature predispose the model to generate maturing networks? Why wouldn't orphan vesicles fuse heterotypically, like other vesicles do?

4) The treatment of spatiotemporal aspects of membrane traffic raised two issues:

The description of virtual stacking (Figure 4) is not at all compelling. What is the justification for modeling vesicle fluxes as attractive springs while modeling compartments as repelling charges? That representation does not seem to correspond to any realistic picture of membrane traffic. This part of the story weakens the paper, and should probably be omitted unless it can be revised in a way that makes sense.

One of the reviewers was quite troubled by the lack of spatiotemporal parameters in your model (transport coefficients, distances, compartment sizes, etc). The other two reviewers were of the opinion that the cytoplasm is well mixed, so it is reasonable to assume that vesicles will reach any potential destination. But to convince the skeptical reviewer and other readers, you should explain why spatiotemporal parameters can be ignored for the purposes of your simulation. In particular, the markers used in your simulation reach a steady-state homeostatic distribution, but is it automatically true that all of the other components, such as lipids, will also undergo balanced flows in a given network? Does your model require the unrealistic assumption that every compartment generates the same number and size of vesicles per unit time? Is it obvious that movement between compartment is fast relative to formation and fusion of vesicles, so that spatial aspects can be neglected?

5) In general, the descriptions and discussion need to be expanded significantly to address the questions that will be raised by cell biologists. Perhaps it would be useful to rewrite the paper with input from a cell/membrane biologist. Here are examples of issues that should be addressed:

The introduction is too generic. It does not even mention the Boolean modeling approach, and does not describe either the power and limitations of this approach or the reasons for choosing it to address this particular topic.

The difference between actual and randomized networks in Figure 2 is confusing, and the significance of Figure 2 is hard to understand. Similarly, in Figure 3, what is the difference between special pairs and random pairs? The parameters and assumptions should be clearly described in terms that will make sense to cell biologists.

The phenomenon of compartments becoming richer in compositional complexity over time has not been described experimentally, and doesn't seem to make biological sense. Can you comment on this discrepancy?

Regarding the maturation chains, one reviewer wrote: "However, these striking observations are not clearly interpretable. The authors should provide much more detail on the characteristics of these chains, on the dynamics of their development, and on the number of the chains that possess these particularly complex features. They should also describe the networks that do not contain maturation chains, which actually represent the majority of the outcomes. They should discuss which parameters lead to realistic maturation chains, and which preclude maturation. What is interesting to me is not the fact that they observe maturation as this outcome is expected, but the fact that the maturation networks are so complex and so similar to the real ones, and so frequent. Again they should thoroughly describe the maturation chains, the alternative outcomes and the conditions where they obtain. Without this description we cannot judge the value of the simulations."

Do the authors wish to argue that their simulations reflect a potentially simpler situation early in evolution, and are therefore not constrained to incorporate features of currently existing trafficking systems? If so, they should make this point explicitly.

[Editors' note: further revisions were requested prior to acceptance, as described below.]

Thank you for resubmitting your work entitled "Stacking the odds for Golgi cisternal maturation" for further consideration at *eLife*. Your revised article has been favorably evaluated by Randy Schekman (Senior editor) and three reviewers, one of whom is a member of our Board of Reviewing Editors.

The manuscript has been improved but there are some remaining issues that need to be addressed before acceptance, as outlined below:

Once again the reviewers had an extended discussion of your manuscript. As before, one of the reviewers remains deeply concerned about the lack of spatiotemporal parameters in the simulations, while the other two reviewers feel that your approach makes sense and yields intriguing insights. After judging these arguments, I am prepared to accept your manuscript if you can address the following remaining issues either by revising the paper or by explaining why a revision is not warranted.

1) Even though the basic patterns of compartmental formation and communication can be described by the Boolean network approach, spatiotemporal parameters are clearly important in real cells to ensure that the membrane traffic machinery is efficient enough to support life. This point should be given further emphasis.

More generally, you should discuss the limitations of the described approach. Does the model make testable, falsifiable predictions? I suspect that this one can be addressed by the presentation. For example, if you predicted that maturation chains could have evolved without specific selection for the transport of large cargo, then the modeling matches this prediction. But are other aspects of membrane traffic not captured by your simple model?

2) The authors should discuss how to arrive at the Boolean network approach from a set of (partial) differential equations (similar to Heinrich/Rapoport's model or starting from overdamped Fokker-Planck equations). In particular: Which rates and transport coefficients are considered large or limiting w.r.t others, i.e. which quasi-equilibria are used for setting up the model from more fundamental approaches?

3) The biological meanings of the propensity parameters for budding and fusion are still unclear. Please provide an explanation. For example, does propensity to fuse reflect the spatial proximity of vesicle donor and acceptor compartments?

4) The assumption that orphan vesicles fuse homotypically continues to be troubling. You may not have fully understood the concern raised during the initial submission, so let's try again.

Vesicles are normally expected to fuse heterotypically. If a vesicle fails to find a target compartment, why should it then fuse homotypically? Why couldn't a vesicle of one type fuse heterotypically with a vesicle of another type? A priori, such heterotypic vesicle fusion may be more likely than homotypic vesicle fusion given that vesicles normally undergo heterotypic fusion with a target compartment.

The concern is that if homotypic vesicle fusion is programmed into the model as the default fallback option, then the model may be biased toward compartment formation and subsequent maturation.

5) Designating a GTPase that cycles through the cytoplasm as being formally equivalent to a vesicle is confusing. If your formalism treats a vesicle and a reversibly associating peripheral membrane protein as being equivalent, you should consider a term other than "vesicle". Maybe "carrier" would be more generic, and "membrane traffic" could be used rather than "vesicle traffic"?

6) At least some lipids can actually move via nonvesicular as well as vesicular pathways. The discussion of lipids may now be overly complex. The bottom line is that some components can only exchange between compartments via membrane vesicles (e.g., transmembrane proteins), some components could exchange either in vesicles or through the cytoplasm (e.g., certain lipids), and some components probably exchange only through the cytoplasm (e.g., certain GTPases).

7) Relevant to #7 above: does the model take into account the fact that the various components traveling from one particular compartment to another could travel at different times and at different rates? Maybe this point is implicit in the model, but it's hard to tell.

8) I didn't understand the discussion in the Results section about large global structures and shuffling the edges of a network. Is this issue important? If so, it should be explained better. If not, it should be omitted.

[Editors' note: further revisions were requested prior to acceptance, as described below.]

Thank you for resubmitting your work entitled "Stacking the odds for Golgi cisternal maturation" for further consideration at *eLife*. Your revised article has been favorably evaluated by Randy Schekman (Senior editor) and a Reviewing editor.

The manuscript has been improved but there are some remaining issues that need to be addressed before acceptance, as outlined below:

This manuscript has been extensively revised and extended, and is now a substantial contribution that will stimulate thinking in the field. I see no benefit in putting it through another round of review because the key concerns have been addressed. But I will ask the authors to address a few very minor comments about the new "Orphan vesicles and homotypic fusion" section.

1) In cells, many compartments fuse homotypically. For clarity, you should state the assumption that compartments cannot fuse heterotypically with one another.

2) The first paragraph in this section is now perhaps more complex than necessary, and is a bit confusing as a result. For example, I have trouble figuring out what this sentence means: "This is rare: in almost all instances we explored, at least some fusion products of orphans were themselves orphans; in the majority of instances even the initial orphans could not fuse with one-another."

3) "one another" is two words.

Please make these changes and submit a final version.

---

## [Author Response]

*1) Your model assumes that compartment identity is determined by transmembrane and luminal components, which are necessarily exchanged by means of transport vesicles. But in reality, components recruited from the cytosol, such as GTPases and adaptors/coats and phosphoinositide modifying enzymes, play an important and perhaps central role in defining compartment identity. In some cases, Rab cascades are thought to regulate compartmental maturation. While your model necessarily is necessarily a simplified representation of reality, overlooking this fundamental property of the endomembrane system seems hard to justify.*

Our Boolean framework in fact does incorporate cytoplasmic molecules such as Rab GTPases. For brevity we did not focus on this aspect in the original submission, and apologize for giving the impression that we have ignored these key components. To explicitly keep track of such players we can interpret any “vesicles” with only a single nonzero molecular component as a single molecule type moving directly through the cytoplasm. We must then add an explicit label to track the flow of membrane lipids on true vesicles. This is of course optional and a matter of interpretation, since in any case we have only considered a subset of all possible molecules in a vesicle traffic system. It would be interesting to ask what proportion of the molecules of a traffic network are transmembrane type (SNARE-like) and what proportion are membrane-associated (Rab-like). This type of analysis can be done within our framework, and would indeed be interesting to do, but takes us beyond the subject of this submission.

Figure 1—figure supplement 1 discusses an example explicitly including Rab GTPases and their regulators. The text has also been modified to emphasize these non-vesicular transport pathways:

Third paragraph of Results section: “N types of membrane-associated, transmembrane or lumenal molecules”

Sixth paragraph of Results section: “Among active labels the Arf- and Rab-family GTPases […] explicitly assigning a label to membrane lipids.”

Sixth paragraph of Results section: “Though GTPases diversified early in eukaryote evolution […] we do not explicitly distinguish between the Rab- centric and SNARE-centric views.”

First paragraph of Methods section: “Vesicles with a single non-zero label may optionally be interpreted as membrane- associated molecules capable of cytoplasmic non-vesicular transport (Figure 1—figure supplement 1).”

*2) The focus of your analysis is on the subset of networks that include maturation chains. But the majority of the networks do not show maturation. The neglect of those other outcomes is surprising. For example, how often is the traditional vesicle shuttle model replicated? What can be learned from analyzing the full set of networks?*

We had analyzed the full set of networks in the original submission, but had only emphasized the key point that cisternal maturation is the single most common type of motif. In this revision we provide much more extensive discussion of the properties of all networks beyond cisternal maturation, and also carry out new analysis. Note that we now discuss all types of networks, including the traditional transport balanced version, as shown in Figure 2. Apart from this we provide several additions.

First, we explain our sampling procedure more carefully (“Sampling homeostatic vesicle traffic networks”). To generate diverse networks, we actually sample by sweeping over parameters corresponding to the number of molecular labels, the number of adaptor/coatomer types, and how permissive vesicles were in selecting cargo and fusion targets. For each parameter combination we get different types of networks, as discussed in Figure 2—figure supplement 1. We can explore how various network properties depend on one another, as we do in Figure 2, and Figure 2—figure supplement 2.

Second, we explicitly separate out the “trivial” networks. In about half our sampled data, the networks end up in a state where each compartment is completely disconnected from all the others. In real cells this would probably correspond to a completely inactive, perhaps dead, cell. Out of 14,809 homeostatic networks, only 8,614 are non-trivial. It is only among the latter that we should look for any structure at all.

Third, we emphasize that our search for network motifs is completely unbiased. In the original submission we had only discussed the three most frequent motifs (present Figure 3).

We now also discuss all 43,700 potential motifs, and the occurrence of the 100 most frequent motifs (Figure 3—figure supplement 1). The main lesson here is the following: there is indeed a great diversity in the structures that we find among our tens of thousands of networks. But half the non-trivial networks (4,024/8,614) contain a maturation chain. The other half contain hundreds of diverse motifs, each occurring infrequently; no single motif is prominent. To emphasize this, in the abstract we now describe the maturation chain as occurring in “the plurality” of traffic networks (borrowing from the political term in which the single largest party wins the vote).

Finally, we emphasize the following: we are not claiming that all vesicle traffic networks contain maturation chains. We did not put this into our framework by hand, and so we naturally see a lot of diversity. However, the maturation chain does occur in about half of all non-trivial networks. This makes it easily “discoverable” by neutral evolutionary search. In the discussion we make the point that even structures that seem likely in complex systems typically require fine-tuning of parameters, whereas cisternal maturation spontaneously arises in half the non-trivial networks.

Figure 2 discuss the properties of different types of networks in our dataset, and show the parameter ranges where cisternal maturation is likely to occur. Figure 3—figure supplement 1 shows the top 10 network motifs. The text has been modified as follows, discussing the properties of all our networks in more detail:

Abstract: “the plurality of networks contain chains of compartments…

Results section, subsection “Sampling homeostatic vesicle traffic networks”: To systematically explore diverse behaviors […] sampled uniformly over these parameters.”

Results section, subsection “Sampling homeostatic vesicle traffic networks”: “90% had 8 or fewer compartments […]”

Results section, subsection “Cell-like networks and vesicle traffic motifs”: “Cell-like networks are most likely to contain maturation chains at low-to-moderate […]”

Results section, subsection “Cell-like networks and vesicle traffic motifs”: “Nearly half the non-trivial homeostatic networks (4,024/8,614) contained a maturation chain of length one or more. We wondered whether the other half were characterized by other distinct features […]”

Figure 3—figure supplement 1 figure legend: “The 15th most common three-compartment motif, occurring in 344 networks, is the maturation cycle (Figure 3—figure supplement 1). This corresponds to a compartment whose composition oscillates periodically. Cycles of various periods occur in about a quarter (2,407/8,614) of all homeostatic networks.”

Results section, subsection “Cell-like networks and vesicle traffic motifs”: “We next checked if our networks contained any large global structures, beyond the scale of local motifs […]”

First paragraph of Discussion section: “For comparison consider a biological system described by ten parameters” “[…] most types of structures are rare and require fine-tuning of parameters.”

*3) The assumption that vesicles unable to find a target undergo homotypic fusion is troubling. This process may reflect the actual behavior of some vesicle types in cells, but why is homotypic fusion the default assumption? Does this feature predispose the model to generate maturing networks? Why wouldn't orphan vesicles fuse heterotypically, like other vesicles do?*

We apologize for not discussing this point in detail in the original submission. The key restrictive assumption we have made is that heterotypic vesicle fusion does not occur. Our Boolean framework can accommodate heterotypic fusion, and it would be interesting to explore the role of such processes further. But it takes us beyond the topic of the present submission. Regarding homotypic fusion, it is also reasonable to ask why we assume all orphan vesicles can homotypically fuse. This is in fact not a restrictive assumption as long as we are only interested in the homeostatic states, as explained in the revised text. We could, if we wish, assume that only certain pre-defined vesicle subsets can undergo homotypic fusion. Once we run to homeostasis, only such orphan vesicles would be present in the final state. If we now explore over all possible vesicle subsets, we get precisely the same set of homeostatic states. Note that the transient period before we approach homeostasis would indeed be affected by this assumption that all orphan vesicles can fuse. We can use the Boolean framework plus additional ingredients involving homotypic or heterotypic fusion to study in detail how a traffic system recovers homeostasis after a perturbation, and this is an interesting future question to ask.

The text has been modified to discuss these issues:

Fourth paragraph of Results section: “Alternatively assuming only a certain subset of vesicles can undergo homotypic fusion (and exploring over all possible subsets) does not change our results: since no state with accumulating un-fused orphan vesicles can be homeostatic, all orphan vesicles in any homeostatic states we find must fall into this subset. Our approach can be extended to allow heterotypic fusion, or to allow vesicles to simultaneously nucleate new compartments and fuse to existing ones. Additional rules are needed to specify when such processes occur, and we do not consider them here.”

*4) The treatment of spatiotemporal aspects of membrane traffic raised two issues:*

*The description of virtual stacking (Figure 4) is not at all compelling. What is the justification for modeling vesicle fluxes as attractive springs while modeling compartments as repelling charges? That representation does not seem to correspond to any realistic picture of membrane traffic. This part of the story weakens the paper, and should probably be omitted unless it can be revised in a way that makes sense.*

We apologize; the brief description of this issue in the original text was confusing. We have modified this section in the revision after getting feedback from colleagues about clarity.

We have first started the discussion by motivating the background on compartment spatial localization. We have removed the confusing term “virtual stacking” and have explained the motivation of this analysis. It is a speculative analysis which is why we left it to the Discussion section. Briefly: Spatial locations of compartments are caused by many layers of biophysical mechanisms. We were speculating about the potential function of the precise locations. In particular, we explored whether any of the results from the theory of logistic networks (warehouses locations connected by trucks) might apply to cellular vesicle traffic networks. In the efficient logistics context, the goal is to spread out the warehouses as evenly as possible while minimizing the distances travelled by trucks. If we imagine compartments to be warehouses and vesicles to be trucks, what would the optimal spatial organization of compartments look like? There is a simple algorithm which produces this optimal configuration, as described in “Methods: Spatial Optimization”. It involves minimizing a certain “energy function” which measures how uniformly compartments/warehouses are distributed and how far vesicles/trucks must travel. The intuition behind why this energy function works is that it resembles a system of repelling charges and attracting springs, but that is merely for the sake explanation, the optimization procedure is the main point. In our original analysis the text misleadingly seemed to read that the “springs” and “charges” were the actual physical causes of the locations, this is not the case. Of course it is complex biophysical regulation which actually places compartments at precise locations.

Whether these locations in fact do optimize efficient traffic is still an open question. All we have shown is that Golgi stacks are consistent with this hypothesis, but this observation certainly does not prove this hypothesis.

Figure 4 of the main text in the original has been moved to Figure 4—figure supplement 1 of the revision. We have expanded the text where this issue is discussed:

Third paragraph of Discussion section: “the spatial locations of compartments are determined by complex and largely unknown biophysical mechanisms [Yadav & Linstedt, 2011].”

Third paragraph of Discussion section: “We explored the hypothesis that a cell is like an efficient city-wide logistics system, in which no part of a city is too far from a warehouse, and warehouses with more traffic between them are placed closer to one another […]”

Third paragraph of Discussion section: “Measurements with repositioned organelles might be used to test these ideas further [van Bergeijk et al., 2015].”

Methods section, subsection: “Spatial organization”: “We want to distribute the compartments of a vesicle traffic network in such a way that all parts of a cell are close to at least one compartment, while the distance travelled by vesicles between compartments is minimized […]”

*One of the reviewers was quite troubled by the lack of spatiotemporal parameters in your model (transport coefficients, distances, compartment sizes, etc). The other two reviewers were of the opinion that the cytoplasm is well mixed, so it is reasonable to assume that vesicles will reach any potential destination. But to convince the skeptical reviewer and other readers, you should explain why spatiotemporal parameters can be ignored for the purposes of your simulation.*

These are important questions, and we will provide a detailed answer here. We also discuss these issues more briefly in the revised text. Most, if not all, previous models of vesicle traffic have attempted to do justice to details such as spatial location, compartment size, reaction kinetics, and so on. In fact we have used such approaches ourselves. Our paper [Ramadas & Thattai, 2013] builds on the classic model of Heinrich and Rapoport, using differential equations and dynamical systems approaches to understand how the number of compartments and their connectivity might be encoded in microscopic molecular parameters.

We tried for a long time unsuccessfully to extend these models to arbitrary network topologies: given the number of parameters, it is technically unfeasible to explore parameter space in an unbiased manner. However, we noticed from our earlier results that the topologies of the networks arising from such complex models were very robustly determined by broad aspects of molecular specificity.

Once the topology was set, the quantitative behavior including rates of cargo flow were determined by more detailed kinetic parameters. This is why we developed the Boolean approach described here. If topology is what one is interested in, the Boolean model provides the same set of topologies are more detailed kinetic models.

Once a topology is obtained form the Boolean approach, the kinetic parameters can be included afterwards. We are currently working on precisely this aspect in an independent project: we are developing a broad mathematical method to put back kinetic parameters onto a given network topology. This turns out to be useful not just because of the Boolean prescription described here, but because cell biological experiments often help us determine network topology (which vesicles flow between which compartments) but not often the kinetics. We will be happy to share with you the results of our project when it is ready. However, it goes beyond the topic of the present submission.

We have modified the introductory text to discuss these issues:

Second paragraph of Results section: “We have shown [Ramadas & Thattai, 2013] that the topological features of a vesicle traffic network – the number of compartments and their connectivity – are robustly determined by molecular specificity alone, while quantitative features such as the rate of flow of cargo depend on additional kinetic parameters. Here we are primarily interested in homeostatic network topologies. We therefore formalize the properties of vesicle traffic systems using a Boolean framework […]”

Second paragraph of Results section: “We also assume the cytoplasm is well mixed, and do not consider spatial location.”

Second paragraph of Discussion section: “This shows that spatial organization is not a pre-requisite for maturation, and supports the idea of a well-mixed cytoplasm.”

*In particular, the markers used in your simulation reach a steady-state homeostatic distribution, but is it automatically true that all of the other components, such as lipids, will also undergo balanced flows in a given network?*

The lipids themselves can be considered as one of the labels in the system. Therefore, if the network is homeostatic, it is balanced in all its markers. However, there is a subtle point that we now provide new analysis to discuss. For each compartment individually, all flows must balance in a homeostatic network. However, is possible that for larger groups of compartments, molecules might flow in but not out (e.g. if two compartments exchange a molecule with one another, but also that molecule is received from a third compartment).

This means that the network is balanced at fast timescales, but on slower timescales this type of molecule will have to be synthesized elsewhere in the system. We now show how often this type of slow synthesis is required, across all the networks in our system. See Figure 2—figure supplement 2.

New text has been added:

Fifth paragraph of Results section: “We can distinguish true vesicles from non-vesicular transport by explicitly assigning a label to membrane lipids […]”

Results section, subsection “Sampling homeostatic vesicle traffic networks”: “In homeostatic networks, molecular loss must be offset by molecular gain at each compartment, and on fast timescales these processes balance out. However, certain compartment groups could receive particular molecules, yet not give them away. When flows are interpreted in terms of continuous fluxes, such molecular sinks imply the need for compensatory synthesis elsewhere on slower timescales. In general, larger networks tended to require synthesis of more types of molecular components (Figure 2—figure supplement 2).”

*Does your model require the unrealistic assumption that every compartment generates the same number and size of vesicles per unit time?*

We do not make this unrealistic assumption. Our model specifies the types of cargo carried by the vesicles generated by each compartment. It is agnostic about the relative amounts of this cargo, about the size of the vesicles, and about the rates of vesicle production. In particular (see our remarks above), once a network topology is provided from our Boolean model, we can then go into detail to examine the molecular flows (e.g. see the new Figure 2—figure supplement 2). Each molecule moves in cycles, and the rate of flux of each cycle can be set as an independent kinetic parameter. We have discussed this in the text, as described above.

New text has been added:

Second paragraph of Results section: “Our approach is agnostic to the relative amounts of molecules on each vesicle, the size of vesicles and compartments, and the flux of vesicles between compartments. We also assume the cytoplasm is well mixed, and do not consider spatial location. Once a homeostatic traffic topology is discovered using this Boolean approach, we can add back more detailed kinetic parameters and examine quantitative molecular flows.”

*Is it obvious that movement between compartment is fast relative to formation and fusion of vesicles, so that spatial aspects can be neglected?*

This is an important point. It is not at all obvious that the spatial configuration of a cell should be neglected. Indeed, it is extremely likely that local concentration effects ensure that vesicles are more likely to find targets closer to the source compartment. We know the cytoplasm does undergo mixing events, partly driven by a dynamic cytoskeleton. However, to our knowledge there is only partial experimental support for the idea of a truly well-mixed cytoplasm.

Our response to this critique is as follows: we are particularly exploring here how far molecular specificities alone can determine the properties of a vesicle traffic system. If these basic ingredients alone are sufficient to generate complex structures, then it is not necessary to invoke spatial organization as an additional input ingredient.

No model is ever complete, the set of ingredients missing in any given model is infinite.

The key question is, do the set of ingredients that are included provide any insight. We hope that our results here cross this bar: that the surprising emergence of cisternal maturation in the absence of spatial organization at least opens up the possibility that maturation is primarily determined by molecular interactions. The recent observation by Glick and colleagues of retrograde vesicle flow in un-stacked Golgi provides some support for this idea.

Figure 2—figure supplement 2 discuss these issues. The text has been expanded in several places to indicate that our Boolean model provides a topological “chassis” on top of which kinetic parameters can be included.

Second paragraph of Results section: “Previous mathematical models of vesicle traffic … add back more detailed kinetic parameters and examine quantitative molecular flows.”

Second paragraph of Results section: “We also assume the cytoplasm is well mixed, and do not consider spatial location.”

Results section, subsection: “Sampling homeostatic vesicle traffic networks”: “In homeostatic networks, molecular loss must be offset by molecular gain at each compartment, and on fast timescales these processes balance out … In general, larger networks tended to require synthesis of more types of molecular components (Figure 2—figure supplement 2).”

Third paragraph of Discussion section: “The connectivity of a vesicle traffic network is primarily determined by molecular specificity, while the spatial locations of compartments are determined by complex and largely unknown biophysical mechanisms [Yadav & Linstedt, 2011].”

Second paragraph of Discussion: “This shows that spatial organization is not a pre-requisite for maturation, and supports the idea of a well-mixed cytoplasm.”

*5) In general, the descriptions and discussion need to be expanded significantly to address the questions that will be raised by cell biologists. Perhaps it would be useful to rewrite the paper with input from a cell/membrane biologist. Here are examples of issues that should be addressed:*

We thank the reviewers for this suggestion, which we have taken to heart. We have circulated the daft among our colleagues in cell biology and have incorporated their suggestions on readability and accessibility. One of the main points we heard was that the model seemed too abstract. To remedy this we have added new text describing how to interpret the molecular labels. We have also added additional clarifying material throughout the text.

New text has been added:

Fifth paragraph of Results section: “This Boolean description appears abstract, but the biology is embodied in the interpretation of the molecular labels. …”

*The introduction is too generic. It does not even mention the Boolean modeling approach, and does not describe either the power and limitations of this approach or the reasons for choosing it to address this particular topic.*

We apologize for the brevity of the discussion in our original submission. This was primarily due to the space constraints of the *eLife* “Short Reports” format.

We have now added an entire paragraph providing background on vesicle traffic models and motivating the use of the Boolean approach. We also direct the reader to our companion manuscript [Mani & Thattai, 2016] which provides an accessible non-mathematical introduction to our Boolean approach.

New text has been added:

Second paragraph of Results section: “Previous mathematical analyses of vesicle traffic have considered such details as compartment size, location and chemical composition, and used ordinary or partial differential equations to study vesicle and compartment dynamics […]”

*The difference between actual and randomized networks in Figure 2 is confusing, and the significance of Figure 2 is hard to understand. Similarly, in Figure 3, what is the difference between special pairs and random pairs? The parameters and assumptions should be clearly described in terms that will make sense to cell biologists.*

We agree this point was confusing in the original submission, and apologize for this. We have significantly expanded our discussion here. To be clear, our analysis proceeded in 3 steps. (1) We explicitly looked for cases of compartmental maturation. (2) We next looked at 3-compartment motifs in an unbiased manner, as discussed above. (3) We finally looked for structures larger than 3-compartment motifs. For step (3), the key to finding larger structures is to shuffle a network while keeping local properties fixed. If this shuffling does not change e.g. the frequency of occurrence of various 3-compartment motifs, then we can be confident that there are no larger repeated structures present. Indeed, this is what we see.

The text has been expanded to discuss this point:

Results section, subsection: “Cell-like networks and vesicle traffic motifs”: “We next checked if our networks contained any large global structures, beyond the scale of local motifs. If such structures are present, they should be disrupted when we shuffle the edges of a network while keeping local incoming and outgoing edges of individual compartments fixed.”

*The phenomenon of compartments becoming richer in compositional complexity over time has not been described experimentally, and doesn't seem to make biological sense. Can you comment on this discrepancy?*

This is an observation we have made from our data, it is not an ingredient we put into the model. The caveat is that we are only modeling a network with a small number of molecular components, and in particular we have focused on the active labels, not passive cargo. In terms of the active labels, imagine there are various Golgi-resident markers constantly treadmilling within the Golgi itself.

What we find is that the very first compartment has fewer active labels than the very last compartment, and this difference causes a gradient within the maturation chain. In a real cell much of the content of the Golgi will be secretory cargo destined for one-way outward flow. These one-way secretions are not considered in our homeostatic framework. Once such passive secretory cargo are introduced, it is less likely that the clear pattern we have observed will remain. In any event, given these various caveats we have opted to remove this particular figure from our revision, and mentioned this result in the text, with the caveat:

Results section, subsection: “Cell-like networks and vesicle traffic motifs”: “The first compartment of a maturation chain tended to be compositionally simpler than its successors (at least in terms of the active labels considered here) due to the treadmilling molecules present on the latter.”

*Regarding the maturation chains, one reviewer wrote: "However, these striking observations are not clearly interpretable. The authors should provide much more detail on the characteristics of these chains, on the dynamics of their development, and on the number of the chains that possess these particularly complex features. They should also describe the networks that do not contain maturation chains, which actually represent the majority of the outcomes. They should discuss which parameters lead to realistic maturation chains, and which preclude maturation..… What is interesting to me is not the fact that they observe maturation as this outcome is expected, but the fact that the maturation networks are so complex and so similar to the real ones, and so frequent. Again they should thoroughly describe the maturation chains, the alternative outcomes and the conditions where they obtain. Without this description we cannot judge the value of the simulations."*

We agree with the reviewer that there is much more to explore in this diverse dataset. Indeed, we hope the approach we have put forward here is used to uncover further reproducible aspects of these networks. In many ways it is the variability and not the common features that might be important: e.g. during cell differentiation, or during pathogenic re-wiring of a host network. However, all these issues take us beyond the present topic. We will pursue them further.

As described above, in our revision we have devoted a great deal of the text to discussing the properties of all 14,809 homeostatic networks in our dataset, not just the ones with maturation chains. In particular, 6,195 are trivial (all compartments are disconnected from one another). Of the remaining 8,614, about half contain maturation chains. We have now provided a lot of information on the half that do not contain maturation chains, in terms of the top 10 motifs (Figure 3—figure supplement 1) as well as other general features (Figure 2 and Figure 2—figure supplement 2). We have shown which parameter ranges produce maturation chains (Figure 2 and Figure 2—figure supplement 2). We have also closed the Results section with a clear statement of our central finding, about the precise number of chains that display all the diagnostic features of cisternal maturation.

Figure 2 and its supplements discuss these issues in detail. New text has been added:

Results section, subsection: “Cell-like networks and vesicle traffic motifs “: “This diverse sample of networks represents a neutral null hypothesis, which can be contrasted with hypotheses about the selection of various vesicle traffic structures. If some structure of interest frequently occurs in our sample, it suggests such a structure might arise non-adaptively.”

Results section, subsection: “Retrograde vesicles emerge spontaneously, driving cisternal maturation”: “Distilling these observations, here is our central result: across 8,614 non-trivial homeostatic networks, we find 3,111 maturation chains which precisely match all three diagnostic features of the cisternal maturation model (Figure 4).”

*Do the authors wish to argue that their simulations reflect a potentially simpler situation early in evolution, and are therefore not constrained to incorporate features of currently existing trafficking systems? If so, they should make this point explicitly.*

This is an interesting point, we thank the reviewer for suggesting it. We have now included a discussion early eukaryote evolution in the introduction:

Fifth paragraph of Results section: “Though GTPases diversified early in eukaryote evolution [Elias et al., 2012], it is plausible that the primordial vesicle traffic system relied on a more basic toolkit of lipidbased compartment identity and SNARE-mediated fusion specificity [Dey et al., 2016].”

[Editors' note: further revisions were requested prior to acceptance, as described below.]

*1) Even though the basic patterns of compartmental formation and communication can be described by the Boolean network approach, spatiotemporal parameters are clearly important in real cells to ensure that the membrane traffic machinery is efficient enough to support life. This point should be given further emphasis.*

*More generally, you should discuss the limitations of the described approach. Does the model make testable, falsifiable predictions? I suspect that this one can be addressed by the presentation. For example, if you predicted that maturation chains could have evolved without specific selection for the transport of large cargo, then the modeling matches this prediction. But are other aspects of membrane traffic not captured by your simple model?*

We have significantly revised the manuscript to emphasize spatiotemporal considerations. The stochastic model (Box) unpacks various mechanistic assumptions in detail, e.g. the cooperative recruitment of vesicles, or the effect of compartment size on vesicle fusion. We have also shown quantitative parametric conditions under which we expect maturation to arise. The detailed stochastic dynamics compare very favorably with the live-cell dynamics of compartmental maturation observed by Losev et al. (Nature, 2006). Throughout the text we have updated the presentation to explain the motivation behind our Boolean framework, and highlighted connections with detailed mechanistic or molecular issues.

*2) The authors should discuss how to arrive at the Boolean network approach from a set of (partial) differential equations (similar to Heinrich/Rapoport's model or starting from overdamped Fokker-Planck equations). In particular: Which rates and transport coefficients are considered large or limiting w.r.t others, i.e. which quasi-equilibria are used for setting up the model from more fundamental approaches?*

We appreciate this question, which many other readers will very likely also have. We have therefore addressed it in full. We have presented (Box) a full microscopic model based on fundamental stochastic processes of vesicle budding and fusion, simulated using Gillespie’s exact stochastic simulation algorithm. The mean-field limit of this simulation produces a set of approximate ordinary differential equations. We have presented the parametric conditions under which this set of equations produces a limit-cycle oscillation, of which the Boolean dynamics are a discrete approximation. This discussion provides readers intuition about the meaning of our abstract vesicle traffic graphs, about the mechanistic underpinnings of our approach, about the timescales involved, etc.

*3) The biological meanings of the propensity parameters for budding and fusion are still unclear. Please provide an explanation. For example, does propensity to fuse reflect the spatial proximity of vesicle donor and acceptor compartments?*

We have clarified the discussion of our propensity parameters. These reflect chemical rather than spatial properties. In brief: we imagine that different species might encode different numbers of e.g. adaptors, Arfs, Rabs, and SNARES. Moreoever, these proteins will also have different chemical interaction specificities. A priori we don’t know whether adaptors tend to select only a small proportion or a large proportion of available cargo. Similarly, we don’t know whether SNARE-SNARE pairings are rather promiscuous, or very specific. The propensities *g* and *f* are varied so we can sample systems of varying degrees of cargo-adaptor specificity and SNARE-SNARE specificity.

Results section, subsection: “Sampling homeostatic vesicle traffic networks”: “These parameters capture genetically-encoded properties such as the number, and the degree of interaction specificity, of cargo, adaptors, Arf and Rab GTPases, and SNAREs.”

Discussion section, subsection: “Sampling homeostatic vesicle traffic networks”: “All these parameters are ultimately genetically determined: N and A represent the number of protein types involved, while g and f summarize their interaction properties. For example, g depends on cargo-adaptor specificity, and f depends on SNARE-SNARE specificity.”

*4) The assumption that orphan vesicles fuse homotypically continues to be troubling. You may not have fully understood the concern raised during the initial submission, so let's try again.*

*Vesicles are normally expected to fuse heterotypically. If a vesicle fails to find a target compartment, why should it then fuse homotypically? Why couldn't a vesicle of one type fuse heterotypically with a vesicle of another type? A priori, such heterotypic vesicle fusion may be more likely than homotypic vesicle fusion given that vesicles normally undergo heterotypic fusion with a target compartment.*

*The concern is that if homotypic vesicle fusion is programmed into the model as the default fallback option, then the model may be biased toward compartment formation and subsequent maturation.*

We have performed new simulations including the possibility of heterotypic fusion of vesicles governed by the same fusion matrix that governs vesicle-compartment fusion. We find that in the majority of instances this does not significantly deplete the set of orphan vesicles, if at all. Moreover, the fusion matrix itself cannot be the mechanism by which small vesicles fuse to form large compartments de novo, in a homeostatic state. The reason is, the desired compartment would already be present, and therefore the vesicle would already fuse to the compartment and not appear as an orphan. We now discuss at multiple places our treatment of homotypic fusion, to clarify our assumptions. We examined the legitimate concern about biasing the model toward having maturation chains. In fact, the bulk of our homeostatic states have very little homotypic fusion. Over a thousand states have no homotypic fusion and still have maturation cycles. We hope this addresses the concern.

Fifth paragraph of Results section: New section: “Orphan vesicles and homotypic fusion.”

Fifth paragraph of Results section: “We explored allowing orphan vesicles to fuse heterotypically according to the same specificity rules that governed vesicle-compartment fusion.”

Seventh paragraph of Results section: “In practice we find that a small dose of homotypic fusion is often sufficient to achieve homeostasis: over 80% of our homeostatic networks require two or fewer types of homotypically fusing vesicles.”

Results section, subsection: “Cell-like networks and vesicle traffic motifs”: “Out of 3,173 non-trivial networks with no homotypic fusion, over a third contained cycles of maturing compartments (1,053/3,173).”

*5) Designating a GTPase that cycles through the cytoplasm as being formally equivalent to a vesicle is confusing. If your formalism treats a vesicle and a reversibly associating peripheral membrane protein as being equivalent, you should consider a term other than "vesicle". Maybe "carrier" would be more generic, and "membrane traffic" could be used rather than "vesicle traffic"?*

We apologize, we have clarified the text to avoid any confusion. We now refer to “vesicles”, or to “non-vesicular carriers”. When there is any scope for ambiguity, we refer to “transport vesicles”. We also clarify the many scales of membrane-bound structures in our model.

Fourth paragraph of Results section: “By explicitly assigning a label to membrane lipids, the Boolean framework can accommodate both vesicular and non-vesicular pathways.”

Results section, subsection: “Orphan vesicles and homotypic fusion”: “We assume a hierarchy of membrane-bound structures: compartments are large; transport vesicles are small; vesicles can fuse to compartments but compartments cannot fuse with one-another.”

*6) At least some lipids can actually move via nonvesicular as well as vesicular pathways. The discussion of lipids may now be overly complex. The bottom line is that some components can only exchange between compartments via membrane vesicles (e.g., transmembrane proteins), some components could exchange either in vesicles or through the cytoplasm (e.g., certain lipids), and some components probably exchange only through the cytoplasm (e.g., certain GTPases).*

We have simplified the text as suggested. We have removed the comments about early eukaryotic evolution since they are peripheral to the main discussion.

*7) Relevant to #7 above: does the model take into account the fact that the various components traveling from one particular compartment to another could travel at different times and at different rates? Maybe this point is implicit in the model, but it's hard to tell.*

We hope the extensive new discussion of stochastic and differential equation models, and the connection of such models with our Boolean framework, clarifies this point.

*8) I didn't understand the discussion in the Results section about large global structures and shuffling the edges of a network. Is this issue important? If so, it should be explained better. If not, it should be omitted.*

We have omitted the discussion about shuffling edges from the main text. We have replaced it with a brief comment that might be relevant to a subset of readers, and directed them to a new reference [Albert & Barabasi, 2002] for further details.

Results section, subsection “Cell-like networks and vesicle traffic motifs”: “This suggests that vesicle traffic networks might usefully be described by the random graph generation approaches that have provided insights into the structure of metabolic, neural, and ecological networks [Albert & Barabási, 2002] (Methods: Randomly shuffled vesicle traffic networks).”

[Editors' note: further revisions were requested prior to acceptance, as described below.]

*1) In cells, many compartments fuse homotypically. For clarity, you should state the assumption that compartments cannot fuse heterotypically with one another.*

We have stated our explicit assumption that compartments cannot fuse heterotypically.

*2) The first paragraph in this section is now perhaps more complex than necessary, and is a bit confusing as a result. For example, I have trouble figuring out what this sentence means: "This is rare: in almost all instances we explored, at least some fusion products of orphans were themselves orphans; in the majority of instances even the initial orphans could not fuse with one-another."*

We have simplified the first paragraph of the section “Orphan vesicles and homotypic fusion” as suggested.

*3) "one another" is two words.*

We have fixed all incorrect instances of the phrase “one another”.